



# NYTEFOX - The NY-Ålesund TurbulencE Fiber Optic eXperiment investigating the Arctic boundary layer, Svalbard

Marie-Louise Zeller[1*], Jannis-Michael Huss[2*], Lena Pfister[2,3], Karl E Lapo[2,4], Daniela Littmann[1], Johann Schneider[2], Alexander Schulz[1], and Christoph K Thomas[2,4]

[1]Alfred Wegener Institute, Helmholtz Centre for Polar and Marine Research, Potsdam, Germany
[2]Micrometeorology Group, University Bayreuth, Bayreuth, Germany
[3]*now at:* Department of Atmospheric and Cryospheric Sciences, University of Innsbruck, Innsbruck, Austria
[4]Bayreuth Center of Ecology and Environmental Research, BayCEER, University Bayreuth, Bayreuth, Germany
[*]These authors contributed equally.

**Correspondence:** Christoph Thomas (Christoph.Thomas@uni-bayreuth.de)

**Abstract.** The NY-Ålesund TurbulencE Fiber Optic eXperiment, NYTEFOX, was a field experiment at the Arctic site Ny-Ålesund (11.9 ° E, 78.9 ° N) and yielded a unique meteorological data set. These data describe the distribution of heat, airflows, and exchange in the Arctic boundary layer for a period of 14 days from 26 February to 10 March 2020. NYTEFOX is the first field experiment to investigate the heterogeneity of airflow and its transport in temperatures, wind, and kinetic energy

in the Arctic environment using the Fiber-Optic Distributed Sensing (FODS) technique for horizontal and vertical observations. FODS air temperature and wind speed were observed at a spatial resolution of $0.127\,\mathrm{m}$ and $9\,\mathrm{s}$ in time along a horizontal array of $700\,\mathrm{m}$ at $1\,\mathrm{m}$ height above ground level (agl) and along three $7\,\mathrm{m}$ vertical profiles. Ancillary data were collected from three sonic anemometers and an acoustic profiler (miniSodar, SOund Detection And Ranging) yielding turbulent flow statistics and vertical profiles in the lowest $300\,\mathrm{m}$ agl, respectively. The observations from this field campaign are publicly available on

Zenodo (DOI: 10.5281/zenodo.4335461) and supplement the data set operationally collected by the Basic Surface Radiation Network (BSRN) meteorological data set at Ny-Ålesund, Svalbard.

## 1 Introduction

Atmospheric model predictions are either established components of our everyday life – such as weather forecasts - or subject

of vivid scientific, political and public discussion when it comes to climate projections.

A key quantity in atmospheric models is the transport of heat, momentum, and matter within and across the atmospheric boundary layer (ABL), whose state is most critical for life on Earth. Despite its essential role in Earth system the behavior of the ABL is poorly understood for large areas and periods where the boundary layer tends to be stably stratified (SBL) and therefore does not follow similarity theories which apply to the convective boundary layer (CBL) (Sun et al., 2020; Thomas,





2011; Sun et al., 2012; Stiperski and Calaf, 2018; Pfister et al., 2021a; Mahrt, 2010; Acevedo et al., 2014). As a consequence, climate predictions in areas systematically prone to SBLs such as polar regions suffer from largest uncertainties for e.g. the 2-m temperature, which is highly affected by SBL processes (Holtslag et al., 2013; Davy and Esau, 2014; Stocker, 2014). Therefore, understanding the underlying mechanisms and forcings of key variables is of utmost importance as the rate of warming affects the Arctic more than twice as fast as global average, a phenomenon commonly known as Arctic Amplification (Cohen et al.,

2014; Overland et al., 2016; Davy and Esau, 2014).

A suitable location for conducting SBL research is Ny-Ålesund, Svalbard, which is a center for several polar research institutions including the AWIPEV station. It hosts several long-term observing systems providing complementary observations. Located at 79°N in the Arctic ocean it experiences long-lived SBLs during the polar night, as well as diurnal SBLs during transition seasons.

Under stable weak-wind conditions classic theories predict turbulence to be totally suppressed by dynamic stability (Monin and Obukhov, 1954). However, a large body of evidence demonstrates that turbulent motions are maintained even for extremely stable conditions (Acevedo et al., 2007; Galperin et al., 2007; Mahrt et al., 2013; Zeeman et al., 2015; Zilitinkevich et al., 2008). This weak-wind turbulence differs greatly from the turbulence dominating the CBL. It covers a broad variety of motions, summarized as submeso-scale motions (e.g. Mahrt et al. (2009)), which do not correspond to any classic similarity assumption

but are significantly non-stationary (Kang et al., 2015; Mahrt et al., 2009).

The fast-evolving, transient, or quasi-stationary nature of submeso-scale motions prompts the development of novel observational systems capable of resolving their temporal and spatial scales: Contrary to classic isotropic and homogeneous turbulence, propagation speed and direction of submeso-scale motions may differ from those of the mean airflow. Taylor's hypothesis of frozen turbulence may not be appropriate to translate temporal observations at one point into spatial scales since ergodicity is

often violated (Mahrt et al., 2009; Thomas, 2011). Therefore, to investigate the behavior and motions of the SBL, real spatial observations on an appropriate scale are required (Mahrt and Thomas, 2016). The innovative Fiber-Optic Distributed Sensing (FODS) technique (Selker et al., 2006a; Thomas et al., 2012; Pfister et al., 2017) offers the much needed observational capabilities and is at the focus of this unique Arctic field campaign. We deployed a large horizontal trapezoidal-shaped FODS array of 700 m length in combination with three vertical profiles at its corners of about 7 m height to record air temperatures and

wind speeds at high temporal (9 s) and spatial (0.127 m) scales. Using a high-resolution coil-wrapped FODS column (Sigmund et al., 2017) air and snow temperatures were recorded along a 2.5 m vertical profile at subcentimeter resolution. To validate the results and place them in a broader context, FODS observations were complemented by measurements from three sonic anemometers at the corners of the FODS array to collect high-frequency wind measurements, as well as an acoustic wind profiler (miniSodar, SOund Detection And Ranging) yielding wind statistics between 10 and 300 m agl.


The main objectives of the campaign were:

– to investigate the spatio-temporal variability of the stable Arctic ABL during the polar night and shed light on the poorly understood physical mechanisms which drive or determine turbulent and submeso-scale motions in the SBL. A deeper understanding will help to find parameters that predict the appearance and character of atmospheric mixing and transport.





– to close the observational gap between point measurements of the operational infrastructure at AWIPEV and to evaluate their representativeness for different incident flow regimes. The FODS setup was designed to allow identifying, characterizing and tracking individual atmospheric turbulent and submeso-scale motions over several hundreds of meters. Deploying the miniSodar allowed to also close the observational gap between ground measurements from flux towers and operational wind LIDAR (LIght Detection And Ranging) observations, which are available for 150 m agl upwards.

– to conduct a pilot feasibility study for the technical setup of a large-scale FODS installation in the extreme environment of the Arctic winter.

## 2  Site description

The experiment was conducted in Ny-Ålesund (78°55′24″ N, 11°55′15″ O), one of the northernmost year-round inhabited settlements in the world, located in the Kongsfjord at the west coast of Svalbard's main island "Spitsbergen" (see Fig. 1). To
the north-east the village is confined by the Fjord, to the south and west by mountains of 500 to almost 800 m agl and several glaciers with snouts towards Ny-Ålesund.

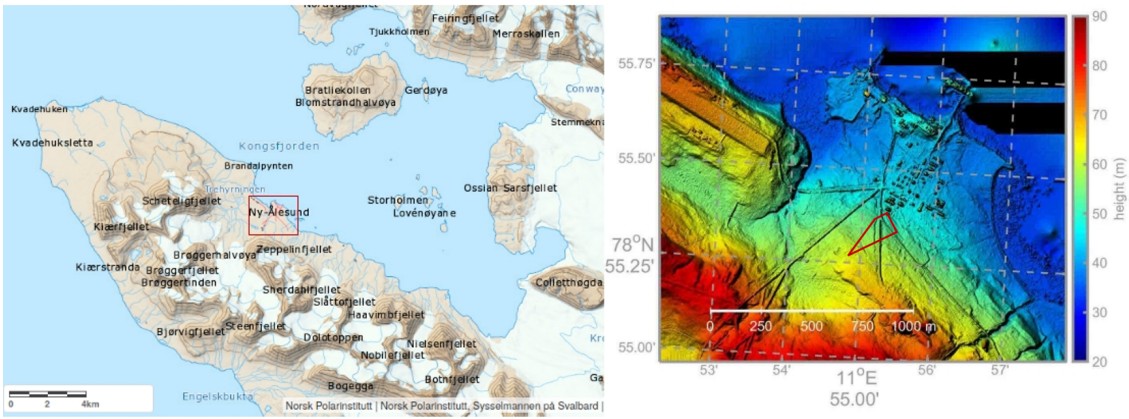

**Figure 1.** Left: Location Ny-Ålesund in the Kongsfjord, retrieved from Norwegian Polar Institute, toposvalbard.npolar.no on 02.12.2020. Right: visualisation of digital elevation model published by Boike et al. (2018) with the setup marked as red shape.

Despite its location at 79°N, Ny-Ålesund experiences relatively mild conditions with mean temperatures varying between $-17.0\,°C$ and $-3.8\,°C$ in January and $4.6\,°C$ and $6.9\,°C$ in July (period of August 1993 to July 2011, Maturilli et al., 2013). These moderate air temperatures are caused by advection of warm air masses from the Atlantic region (Shears et al., 1998)
and due to the West Spitsbergen Current transporting water from the North Atlantic into the Arctic Ocean, passing Svalbard's west coast (Aagaard and Greisman, 1975; Haugan, 1999). However, during the measurement period in February and March 2020, Ny-Ålesund experienced very low temperatures down to $-30\,°C$ with a mean temperature of $-17\,°C$ at 2 m height for the measurement period.





The climate of Ny-Ålesund is strongly influenced by polar night and day, lasting from 24 October to 18 February and from
18 April to 24 August, respectively (Maturilli et al., 2013). Due to the low solar elevation angle in spring, the mountain ridge
south of Ny-Ålesund cast a shadow on the experimental area during the whole field campaign except for very short periods of
direct solar radiation in the last days.

The ABL over Ny-Ålesund is determined by the presence of land-sea contrast, channeling effects induced by the fjord and
the topography, and katabatic airflows from mountains and glaciers in the vicinity of the village. The local wind field is driven
by orography resulting in three main wind sectors: The year-round predominant wind directions are south-east and north-west
corresponding to the fjord axis with the full range of wind speeds (Maturilli et al., 2013; Jocher et al., 2012; Esau and Repina,
2012, Fig. 1). High wind speeds along this axis result from strong synoptical forcing. The third main wind direction is south-
west with wind speeds typically less than $5\,\mathrm{m\,s^{-1}}$ (Maturilli et al., 2013). Southwesterly winds are associated with katabatic
flows down the Zeppelin mountain and the Brøgger glacier and orographic channeling of the flow by the Brøgger massif
(Schulz, 2017). In wintertime, southwesterly winds are often accompanied by stable stratification and gravity waves excited at
low wind speeds (Jocher et al., 2012).

## 3 Setup

The NYTEFOX experiment was conducted at the southern perimeter of the science station Ny-Ålesund. A picture of the field
installation and the settlement taken from the Zeppelin station ($474\,\mathrm{m}$ above sea level (asl)) and the schematic setup is shown in
Figure 2. The setup consisted of six main components, which are displayed in Figure 3, and combined three different sampling
techniques: 1. Fiber-optic distributed sensing including a horizontal array, and vertical low- and high-resolution profiles yield-
ing air and snow temperature and wind speed (Fig. 3 1.A, 1.B, 1.C), 2. ultrasonic anemometers enabling the computation of
atmospheric flux densities using the eddy covariance technique and other flow statistics (Fig. 3 2.), and 3. acoustic ground-based
remote sensing (miniSodar, SOund Detection And Ranging) yielding profile measurements of wind speed and direction and
turbulent mixing strength (Fig. 3 3.). The operational parameters for all sampling systems are listed in Table 1. The horizontal
trapezoidal fiber-optic array had a perimeter of approximately $700\,\mathrm{m}$, whose corners were marked by three $10\,\mathrm{m}$-tall towers
and the AWIPEV Balloon House. One tower was located near the Balloon House and the AWIPEV observatory (referred to
as "tower Obse" marked 'd' in Fig. 3), the second tower was located in close proximity to the AWI eddy covariance station
(referred to as "tower Eddy" marked 'e' in Fig. 3), and the third south of the AWI meteorological tower at the BSRN field
(referred to as "tower BSRN" marked 'i' in Fig. 3).

Ancillary atmospheric observations of the AWIPEV station complement the above mentioned experiment-specific observa-
tional systems. These data include meteorological tower measurements from the Baseline Surface Radiation Network (BSRN)
site (Maturilli, 2020a), as well as balloon-borne meteorological data from radiosondes (Maturilli, 2020b).



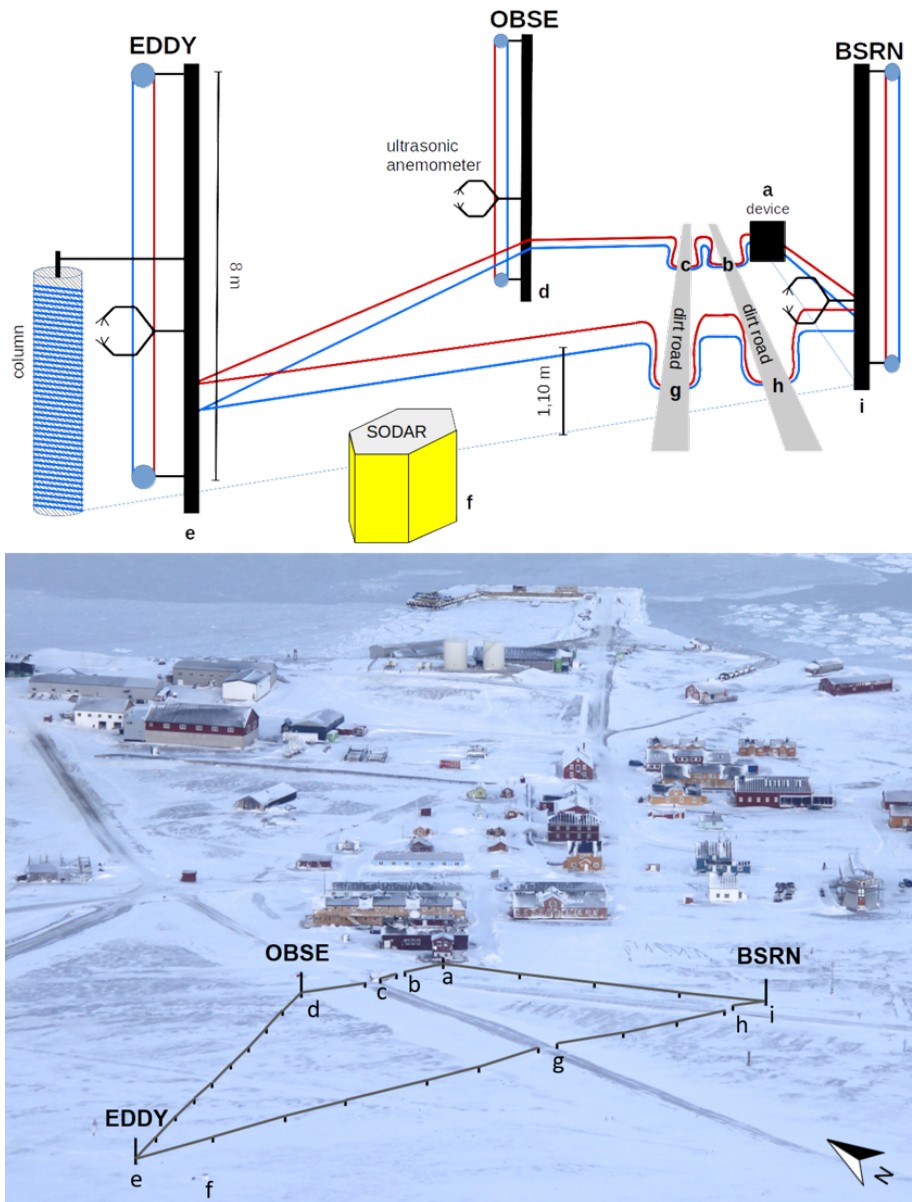

**Figure 2.** Schematic setup and picture of setup from the Zeppelin mountain from south. The fiber-optic array has a length of 700 m. The letters a-i refer to the same elements.

## 3.1 Fibre-Optic Distributed Sensing measurements

The Fibre-Optic Distributed Sensing (Thomas and Selker, 2021) technique can be utilized to measure the spatial and temporal variability of air temperature and wind speed with high spatiotemporal resolution. It enables resolving short-lived turbulent



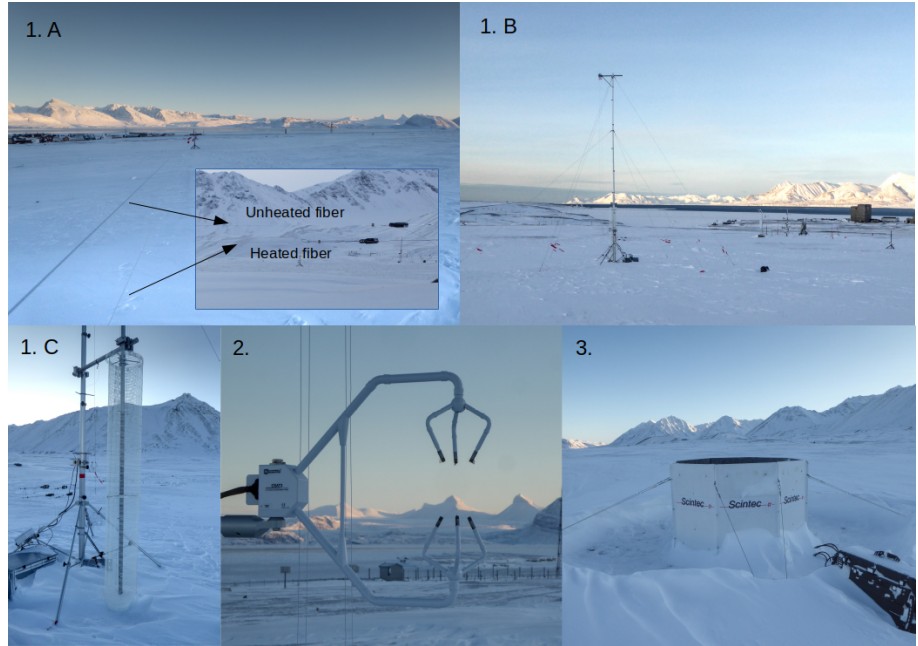

**Figure 3.** Main components of the NYTEFOX field setup: 1.A Fiber-optic cable (metal encased): horizontal temperature and wind speed measurements, 1.B Fiber-optic cable (metal encased): low-resolution vertical profiles of temperature and wind speed across 7 m, here at the Eddy tower, 1.C Fiber-optic cable (PVC-coated): high-resolution vertical temperature profile across the first 2.5 m agl (column), 2. ultrasonic anemometer: flux densities, wind direction and wind speed measurements at the towers, here at the BSRN tower, 3. acoustic profiler (miniSodar): wind measurements located near the Eddy tower.

**Table 1.** Technical data of the used measurement techniques: raw frequency of the devices, temporal resolution and average and spatial average of the measurements.

| measurement | orientation | raw frequency | temporal resolution | temporal average | spatial average |
|---|---|---|---|---|---|
| metal encased fiber | horiz/vert | 3 s | 9 s | 6 s | 0.127 m |
| PVC fiber | vertical | 3 s | 9 s | 3 s | 0.30 m |
| CSat 3 | - | 20 Hz (60 internal) | - | 120 s & 10 min | 0.116 m |
| miniSodar | vertical | 10 s | - | 10 min | 5.0 m |

and longer submeso-scale motions in space and time (Peltola et al., 2020; Thomas et al., 2012; Pfister et al., 2019; Zeeman et al., 2015). A main advantage of FODS is that it does not require assumptions of spatial homogeneity and ergodicity as it explicitly resolves thermal and dynamic structures in space and time (Mahrt et al., 2020; Pfister et al., 2021a, b; Zeeman et al., 2015). Therefore, it is a key technology for investigating spatiotemporal phenomena which cannot be observed by traditional meteorological point measurements or their relatively sparse networks.





The sampling principle of the deployed FODS technique is based upon Raman backscattering (for details see Selker et al. (2006)). The frequency-shifted backscatter of a near-infrared laser pulse emitted into a fiber-optic glass core is analyzed for two spectra bands known as Stokes (red-shifted) and Anti-Stokes (blue-shifted). The ratio of their backscatter intensities is

proportional to the temperature of the light-scattering portion of the fiber-optic cable, which is why this technique is more commonly referred to as distributed temperature sensing (DTS).

In our setup the fiber-optic cable was assumed to be in thermal equilibrium with the air and snow temperatures, which is a reasonable assumption for the low solar-intensity environment of the polar night which helps minimizing the radiative error (Sigmund et al., 2017). Distance along the fiber-optic cable is resolved by range-gating knowing the speed of light and the

length and geometry of the fiber-optic cable. This yields a resolution of $0.127\,\mathrm{m}$ along the cable (Tab. 1) using the highest-resolution DTS device currently on the market (Model Ultima DTS 5km variant, Silixa, London, United Kingdom).

### 3.1.1    FODS reference baths

Since DTS devices yield only relative temperature measurements, portions of the fiber-optic cables are guided through known and stabilized temperature environments, so-called reference calibration sections, to convert the raw Stokes/Anti-stokes ratios

into physically meaningful environmental temperatures (Hausner et al., 2011; Van De Giesen et al., 2012; des Tombe et al., 2020a). Typically, liquid water baths are used as reference sections in which the fiber-optic cable is loosely coiled while stratification is prevented by mechanical mixing. Since liquid water baths are difficult to maintain in the cold Arctic polar night, we deployed a pair of novel solid state reference baths. Each solid state reference bath consisted of a $25\,\mathrm{kg}$ cylinder of pure copper consisting of four interlocking parts. Their design allowed for an internal groove between a central core and an outer ring to

contain several coils of each fiber-optic cable. The temperature of each copper cylinder was controlled thermoeelectrically by Peltier elements to within $\pm\,0.06\,\mathrm{K}$ and observed with two independent high-accuracy platinum resistance (PT-100) thermometers embedded within the copper body next to the fiber-optic cables. The walls of the internal groove housing the fiber-optic cables were painted with a high-emmissivity paint ($\epsilon = 0.95$) to enhance the radiative transfer between the adjacent solid state reference parts to eliminate thermal differences. Each solid state reference bath was contained in an insulated portable case to

minimize temperature fluctuations in time and across the copper core. One solid state reference bath was cooled (referred to as 'cold bath') while the other was heated (referred to as 'warm bath') to span the range of environmental temperatures observed within the fiber-optic array.

Additionally, an ambient (non temperature-controlled) reference bath was deployed at the Eddy tower using an insulated plastic case, whose temperature was measured by a high-precision and accuracy resistance thermometer (RBRsolo[3] T, RBR,

Ottawa, ON, Canada). This bath served as an additional reference section at the far end of the PVC-coated fiber (high-resolution vertical profile) only.



### 3.1.2 FODS measurement components

The key FODS sensor is a pair of two fiber-optic cables, both metal encased loosely-buffered $50\,\mu m$ single-core fiber (outer
diameter $1.12\,mm$, Model C-Tube, Brugg, Switzerland) coated by a $0.2\,mm$ thick polyethylene (PE) white jacket for electric
insulation: one actively heated (red fiber in Fig. 2) and one unheated (blue fiber in Fig. 2) fiber-optical cable to obtain wind
speed measurements in addition to those of air temperature. The underlying principle of wind speed measurements is the
changing temperature difference between both fibers due to convective cooling of the heated cable (see Sayde et al., 2015; van
Ramshorst et al., 2020, for details). Since this cooling is sensitive to the angle of attack, solely winds orthogonal to the fiber are
represented correctly. Horizontally, we therefore only obtain relative wind speed information while vertically this constraint
disappears. Four equally long section were continuously heated in parallel at a variable heating rate adjusted to environmental
conditions (Model Heat Pulse System, Silixa, London, United Kingdom).

     The horizontal fiber-optic array was arranged in a trapezoidal shape (see Fig. 2 and 3 1.A). The fiber-optic cables were
installed at $1.2\,m$ agl with height being measured at the center between fibers and varying with orography. We chose a two-
dimensional array geometry in order to detect propagation of turbulent and submeso-scale structures in all horizontal directions.
This was motivated by the frequently changing and meandering wind directions, known for weak wind conditions. The un-
heated fiber was mounted above the heated at a vertical distance of $0.1\,m$. Every $\approx 30\,m$ tripods were used to support the fiber
to avoid excessive sagging. To keep tension on the fiber, clamping fixtures commonly used for pasture fences (see Fig. 3 1.A)
were mounted at both ends of each section and readjusted when needed. These tensioners were both efficient and relatively
easy to deploy in the extreme Arctic conditions.

     Using $10\,m$-tall towers, vertical fiber-optic profile observations of air temperature and wind speed (referred as low-resolution
vertical profiles, see Fig. 3 1.B) were mounted at three corners of the horizontal array. A total of four fiber-optic sections (heated
and unheated, for each upward and downward direction) were secured by plastic discs at the top and at bottom by horizontal
support booms. Due to radiative and mechanical artifacts induced by these support structures, the effective height of the vertical
profile reduces to around $7\,m$ (see chapter 4.1 for the data processing).

     The third fiber (see Fig. 3 1.C) in the array was an unheated PVC-coated Kevlar reinforced tightly-buffered $50\,\mu m$ single-
core fiber (AFL, Duncan, SC, USA). At the Eddy tower (schematic setup Fig. 2 e) a high-resolution vertical profile consisting
of a $2.5\,m$ height column was used to sample snow and air temperature. The PVC fiber-optic cable was helically coiled around
a support structure made from reinforcement fabric (Sigmund et al., 2017), resulting in a subcentimeter vertical resolution (see
data description in chapter 4.1).

### 3.2 Ultrasonic anemometers measurements

Three ultrasonic anemometers (model CSAT3, Campbell Scientific, Inc.) were installed at each $10\,m$ tower at approximately
$1.4$ to $1.5\,m$ agl and an azimuth angle of about $205°$ to measure turbulent 3-dimensional wind speed components and sonic
(acoustic) temperature at a sampling frequency of $20\,Hz$ (see Fig. 3, 2.). For further details on the measurement technique see
Aubinet et al. (2012) and Foken and Napo (2008).

### 3.3 MiniSodar

To observe airflow across the near-surface and the lower boundary layer a heated ground-based acoustic remote sensing instrument (miniSodar, SOund Detection And Ranging, model SFAS, Scintec AG, Rottenburg, Germany) was setup south of the Eddy tower with an azimuthal orientation of 356° (schematic setup Fig. 2 f, Fig. 3 3.) to measure horizontal wind speed
and direction, vertical velocity variance, backscatter intensity, and turbulence kinertic energy from $10$ up to $300\,\mathrm{m}$ altitude at a $5\,\mathrm{m}$ vertical gate resolution (for further details of the basic operation principle see Neff (1975)). The miniSodar was operated in multi-frequency mode using eight different acoustic frequencies ranging from $2.4$ to $4.8\,\mathrm{kHz}$ outputting averaged Doppler and non-Doppler quantities over $10\,\mathrm{min}$ increments. The miniSodar wind profile complements the existing AWI wind LIDAR (LIght Detection And Ranging) system installed on the observatory roof whose profile observations start at approximately
$150\,\mathrm{m}$ agl.

## 4   Data description

In the following, the data processing procedure for each observational system is presented. Observations from all systems are displayed for a 24-hour period on 5 March 2020, as this day featured a transition between atmospheric flow and temperature regimes in the early afternoon hours.

### 4.1   Fiber Optic Distributed Sensing

The metal-encased fiber (see Fig. 3 1.A and 1.B) was attached to the DTS machine in a double-ended configuration to two different optical channels (Thomas and Selker, 2021) such that the observations from the alternating directions were recorded separately. The unheated and heated fibers of the horizontal array, including the low-resolution vertical profiles, were sampled as one optical path by connecting them via a fusion splice in the middle. We recall that each section within this one optical
path was routed through the solid state reference baths, resulting in a total of eight calibration reference sections.

The PVC-coated fiber (high-resolution vertical profile, Fig. 3 1.C), being operated in a single-ended configuration, was calibrated separately, using the two solid state reference baths and the additional ambient bath at the Eddy tower.

### 4.1.1   Processing steps

The data processing and fiber calibration was done using the software package 'pyfocs', an open-source python library from
the University of Bayreuth Micrometeorology group (Lapo and Freundorfer, 2020). The implemented double-ended calibration procedure is based on des Tombe et al. (2020b).

The FODS data were converted from length along the fiber (LAF) to a geographically-referenced coordinate system. For retrieving this information, several steps had to be performed during and after the measuring period: First, physical locations of all points of interest (e.g. start and end of each defined fiber section) were mapped during the field campaign. Second,
artifacts of the fiber holders, street crossings and edge effects in the calibration sections were removed, employing diagnostics.



Artifacts were visible as spatial perturbations in the mean temperature where the fiber was in contact to solid structures, like the fiber holders, due to different heating or cooling from radiation and/or convection. Additionally, these structures subdue the variability in air temperature and hence diminish the standard deviation of temperature for adjoining fibers. In an iterative process, the section margins were manually adjusted, discarding as little FODS data as possible. Third, all unheated and
heated fiber sections were spatially aligned by finding the maximum spatial cross correlation when no heating was applied. The necessity arises from the wind speed derivation requiring the paired fibers to be spatially aligned. Due to strong vertical gradients, even small mismatches in the vertical coordinate on the order of a single LAF bin strongly reduces data quality. Fourth, the aligned temperatures were mapped to physical geographic coordinates (UTM, with the z-coordinate being height agl) by interpolating the values obtained for the start and end points of each fiber-optic section. As a last step, data were
temporally resampled to an evenly spaced time step of nine seconds to eliminate small deviations in integration times by the internal signal processing in the DTS device.

### 4.1.2 Corrections

Due to deployment-specific technical difficulties in the cold solid state calibration bath, its temperature slowly drifted over time, rendering FODS observations implausible whenever the temperature differences between warm and cold baths were small or
even reversed. Hence, a criterion for quality control was established: Two-minute temporal averages of the sonic temperature were compared to the most closely collocated FODS section in the three vertical low-resolution profiles. Sonic temperatures were converted into dry bulb temperatures, using slow response humidity data from the Baseline Surface Radiation Network (Maturilli et al., 2013). The fiber temperature was spatially averaged over $1\,\mathrm{m}$ centered at the ultrasonic anemometer mounting heights for each profile for the ascending and descending branches of the unheated fiber, resulting in a spatial average over 14
bins. Next, the first approximate derivative of the temperature difference (the change of difference between each two minute interval) between the FODS and sonic temperatures was calculated and averaged across all three towers. All data exceeding $|0.61|$ K per $2\,\mathrm{min}$, defining the upper $99^{th}$ percentile, were rejected. To avoid small data snippets, data between the resulting gaps were rejected if they were shorter than 1h 17min, which was minimum duration needed for scientific interpretation in subsequent data analysis. A total of 20h 50min of data were excluded from the fiber-optic data set for both the metal-encased
and PVC-coated fiber).

The bottom of the high-resolution profile (column, PVC-coated fiber) was immersed in snow with varying density because of uneven snow drift and compaction, resulting in substantial horizontal heterogeneity across the cross section of the column. The heterogeneity manifested itself as systematic alternating stripes of warmer and colder temperatures across each coil.
To eliminate this artifact in snow temperatures, only the side of the column was retained whose signal was most strongly decorrelated with air temperatures above the snow surface. This step led to an effective coarser vertical resolution in the snow of $10\,\mathrm{mm}$ instead of $2.5\,\mathrm{mm}$ in the lowermost aerial section.



### 4.1.3 Final data

All provided fiber-optic data have a temporal resolution of $9\,\mathrm{s}$. The sampling resolution is $0.127\,\mathrm{m}$ for the metal-encased fiber.
The sampling resolution of the PVC fiber used for the high-resolution column was $0.254\,\mathrm{m}$ along the fiber, but the coil wrapping yielded a much higher effective vertical resolution of $2.5\,\mathrm{mm}$ vertically in the lowest quarter (0 to $0.625\,\mathrm{m}$), $5\,\mathrm{mm}$ from $0.625$ to $1.25\,\mathrm{m}$, $10\,\mathrm{mm}$ from $1.25$ to $1.875\,\mathrm{m}$, and $20\,\mathrm{mm}$ from $1.875$ to $2.5\,\mathrm{m}$. Effective vertical resolution was varied due to the logarithmic nature of vertical temperature and wind speed gradient close to the surface.

In the following we present the observations for the 5 March 2020 as an example for the FODS temperature between the Obse and Eddy tower (Fig. 4). The upper graph (a) displays the entire 24-hour period, covering a distinct temperature regime change around 15:00 UTC. The two temporal subsets below illustrate the different character of structures during a strong wind (b) and weak wind (c) regime, showing the unique capabilities of true spatiotemporal observations. As the different regimes go along with characteristic wind direction patterns, the direction at both towers is plotted additionally in Figure 4.

During the morning temperatures were mostly uniform in space and time caused by the intense shear-driven mixing as a result of high wind speeds (compare Fig. 7). After the breakdown of the strong winds around 13:15 UTC and first meandering at this time it takes almost two hours for the temperature to drop and characteristic weak wind, submeso-scale structures to dominate, such as oscillating wind direction and strong temperature instationarities and gradients, starting around 15 UTC.

The observed oscillations in both speed and direction are a typical submeso-scale phenomenon called meandering (Anfossi
et al., 2005; Mahrt et al., 2009). Especially the wind direction shift from southwest to eastnortheast between 22:00 and 22:30 UTC, which exceeded $180\,^{\circ}$ in magnitude caused the passage of cold-air structures. The dramatic temperature drop by almost $10\,\mathrm{K}$ during its passage suggests that it was katabatic outflow originating from the Brøgger glacier situated southwest of the measurement site.

The visualization proofs that submeso-scale motions during weak wind situations can be resolved and tracked with FODS,
as aimed for in the second objective.

The distributed wind speed observations for the above mentioned temporal weak-wind subset (Fig. 5) indicate a coherence between temperature and wind patterns. While low temperatures mostly go along with low orthogonal wind speeds (compare to Fig 4 c), this implies either overall low velocities or a change of wind direction due to the directional dependence of the
measurement technique (see Chapter 3.1.2; Pfister et al., 2019; van Ramshorst et al., 2020). Overall, variability is higher in time than in space. However, there are still periods where winds varied spatially across the displayed section like e.g. the passing atmospheric feature around 23:30 UTC.

The regime change between a strong wind and a weak wind regime observed horizontally in Figure 4 was also clearly
articulated in the vertical FODS profiles (Figure 6). This change caused an abrupt transition from isothermal to strongly stably conditions, with a surface-based inversion that is captured especially by the high-resolution vertical column (right graph). Note

Earth System
Open Access Science
Data Discussions

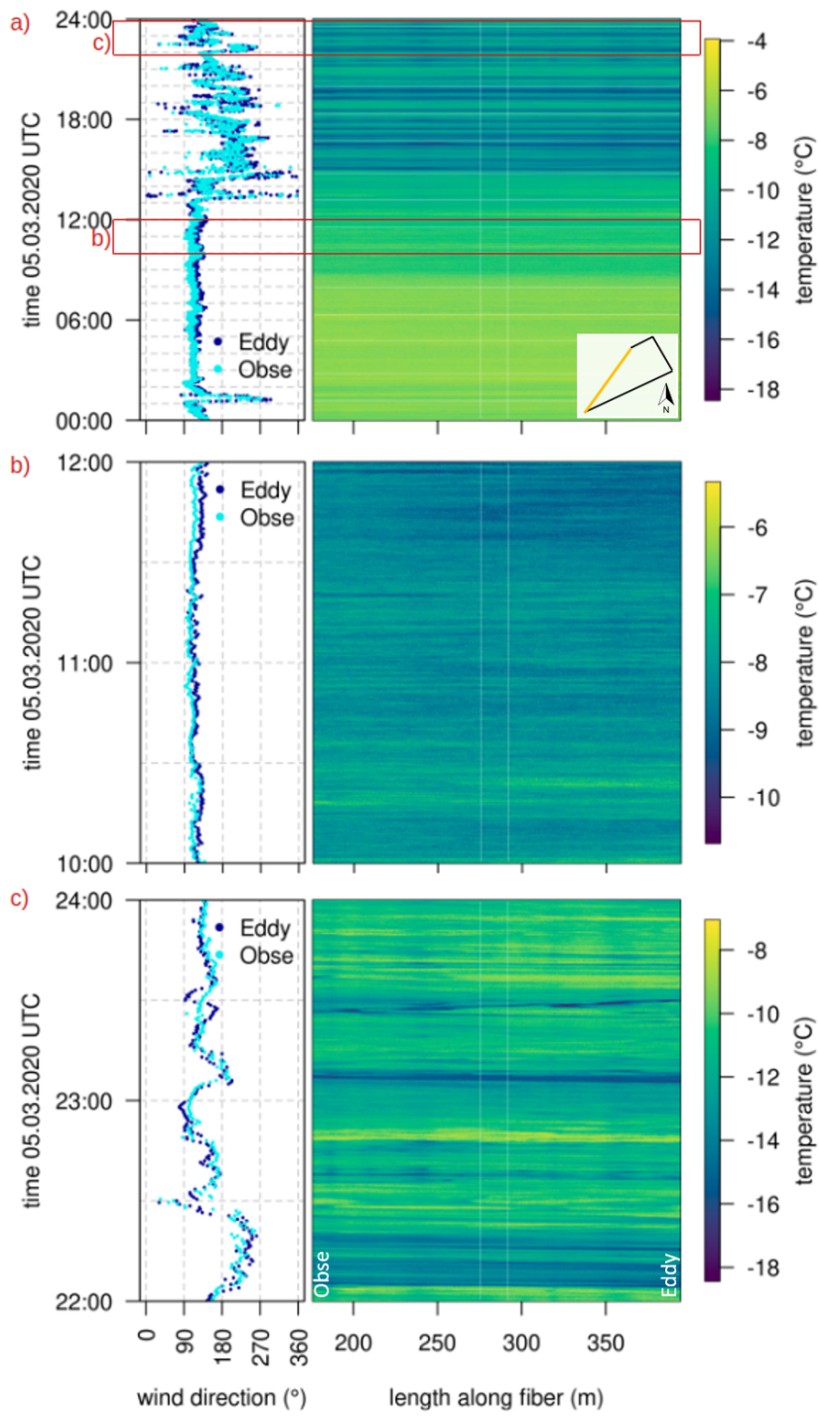

**Figure 4.** Fiber optic temperature along the Obse-Eddy transect and ultrasonic anemometer wind direction (temporal resolution = 30 s) at Obse and Eddy tower; for the whole day, 5th of March (a) and two two-hour subsets during the strong wind regime in the morning (b) and weak wind regime at night (c).
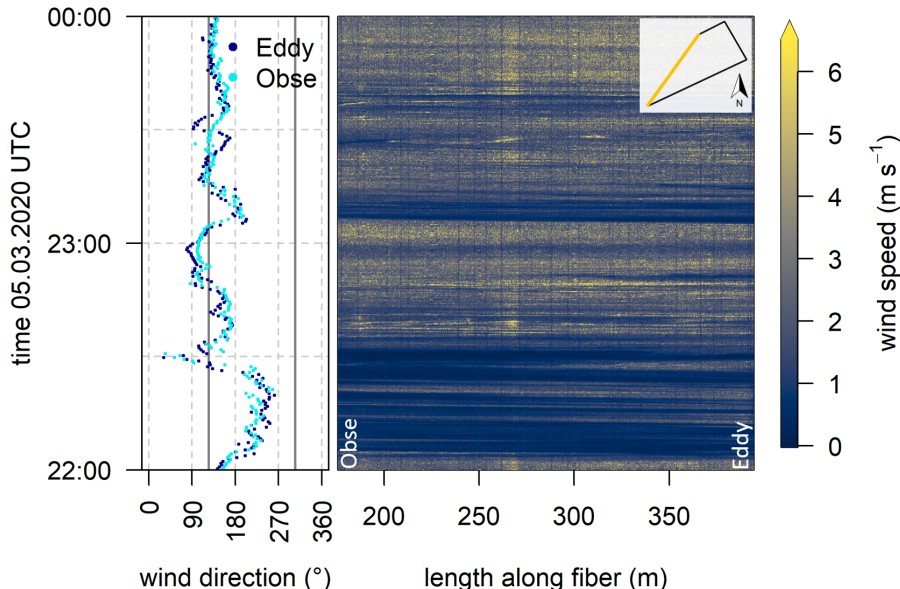

**Figure 5.** Fiber optic wind speed along the Obse-Eddy transect for the 5th of March. For visualisation purposes, some of the stronger artefacts created by fiber holders and marking tape (appearing as vertical lines) were removed by interpolating spatially. Heating rate for this period: $1.6 \, \mathrm{W m^{-1}}$. On the left, ultrasonic wind direction (temporal resolution = 30 s) at Obse and Eddy tower are plotted. Grey vertical lines indicate the wind directions orthogonal to the fiber, where wind speeds are not biased by the angular dependence of the method.

that the lower $0.23 \, \mathrm{m}$ of the column were immersed in snow, which results in higher and more homogeneous temperature values.

### 4.2 Ultrasonic anemometers

Eddy-covariance fluxes from sonic anemometers located on the three towers were computed using a fixed perturbation timescales of 30 s and subsequently averaged to 2 min using the software tool 'bmmflux' of the Micrometeorology Group of the University of Bayreuth (see Appendix in Thomas et al. (2009)). First, the raw data were filtered according to instrument flags and plausibility limits. Subsequently, unphysical turbulence data were removed using a despiking routine (Vickers and Mahrt, 1997). A three-dimensional rotation routine was applied aligning the flow for each averaging interval into the horizontal along- and

cross-wind components and eliminating the mean vertical component potentially caused by a tilt in the ultrasonic anemometer, surface conditions, or semi-stationary eddies of timescales exceeding the perturbation timescale (Wilczak et al., 2001). Computed fluxes were corrected for low- and high-pass losses following Moore (1986). The buoyancy flux was converted into sensible heat flux by a post-hoc buoyancy correction (Liu et al., 2001). Quality flags for turbulent fluxes were computed according to Foken et al. (2004) and applied to discard data either not satisfying stationarity or compliance with similarity theory.

The scheme runs from 1 (best quality) to 2 (worst quality) and we discarded data with flags > 1. For the 2 min data triple or-



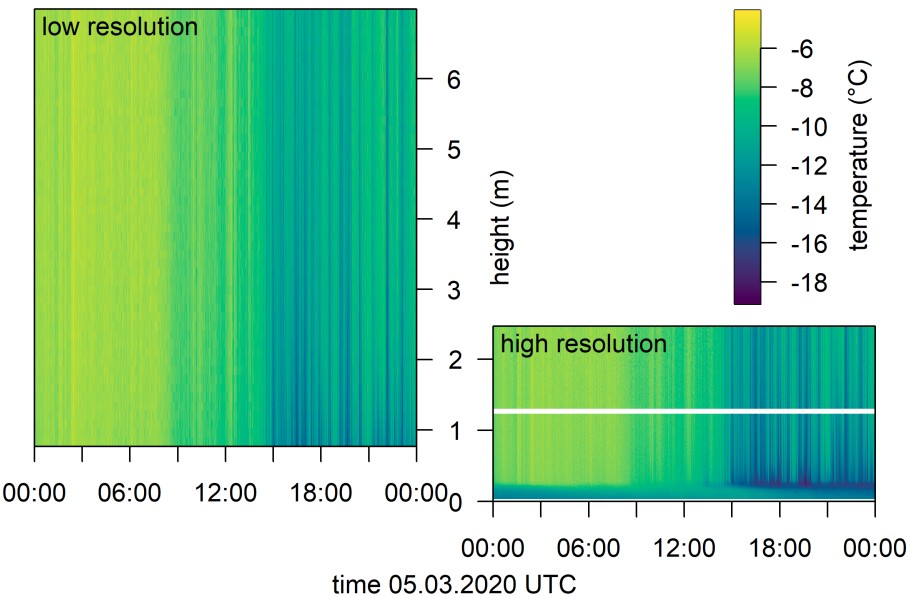

**Figure 6.** Low-resolution vertical temperature profile at Eddy tower (left graph) and high-resolution temperature profile across the first $2.5\,\mathrm{m}$ agl (column) at EDDY tower (right graph) for the 5th of March 2020. The white stripe in the right plot results from rejected data where the fiber was in contact to a plastic support ring.

der moment variables are computed. A comprehensive list of bmmflux output statistics is provided as part of the data repository.

The ultrasonic anemometer wind speed and wind direction are consistent with the regime change observed in the FODS data (Fig. 7): the near-surface easterly airflow starts to decrease in strength around noon, reaching a first minimum of $\leq 1\,\mathrm{ms}^{-1}$ around 13:30. The calmer winds with speeds ranging from $0.5$ to $4\,\mathrm{ms}^{-1}$ came predominantly from southwest, interrupted by sudden distinct wind direction shifts.

### 4.3 Acoustic profiler, miniSodar

Raw acoustic backscatter at all eight frequencies from each acoustic pulse were subject to quality control using the built-in instrument filtering routines for spectral width, ground clutter, signal-to-noise ratio, and plausibility limits. Quality-filtered data from all frequencies were then combined to compute vertical profiles of the horizontal wind speed and direction, vertical velocity variance, turbulence kinetic energy, and backscatter intensity over an averaging interval of $600\,\mathrm{s}$ (Fig. 8).

Identical to the changes found in the FODS and sonic anemometer measurements, the regime shift in wind speed and direction was also observed by the acoustic profiler. However, the profiler observations limit the distinct regime change to southwesterly flows after 13:30 to a maximum depth of $\leq 80\,\mathrm{m}$ agl. Flow further aloft remained to be easterly showing common boundary-layer profiles with a logarithmic increase in wind speeds of up to $\approx 8\,\mathrm{ms}^{-1}$. This supposedly larger-scale

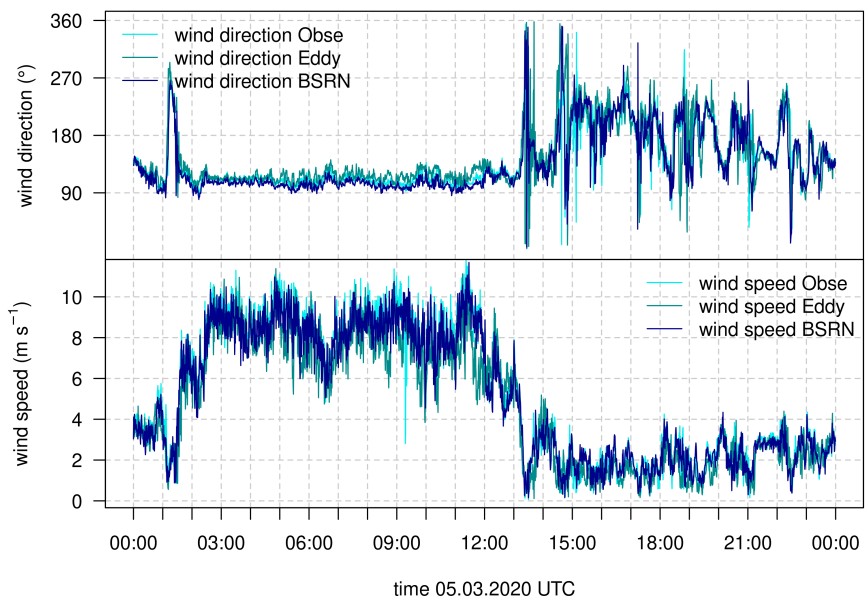

**Figure 7.** Wind direction and wind speed (temporal average 30 s) for all three ultrasonic anemometers at the three towers (Obse, Eddy, BSRN) for the 5th of March.

synoptic flow was interrupted by shorter, approximately 1-hour long periods of very weak westerly winds beyond 140 m agl. The vertical directional shear characteristic for these interruptions suggests strong vertical decoupling and strongly stable near-surface temperature gradients required to maintain the decoupling in spite of relatively strong southwesterly surface winds of up to $4\,\mathrm{m\,s^{-1}}$ (compare Fig. 7). The depth of the katabatic cold-air intrusion from the Brøgger glacier between 22:00 and 22:30 UTC was restricted to the lowest 30 m agl and characterized by a calm period throughout the observation layer. This example period emphasizes the role of local topography as source areas for local flows and submeso-scale motions when the synoptic forcing is negligible. Deriving such potential drivers of submeso-scale motions addresses the first objective of the experiment. The varying maximum measuring height (missing data displayed in grey) was caused by low clouds, snowfall, and wind noise around the acoustic enclosure during the strong-wind period.

# 5 Summary and Outlook

The NYTEFOX field campaign yielded a unique near-surface and boundary-layer meteorological data set for the Arctic polar night above land, which for the first time combines observations from fiber-optic distributed temperature and wind sensing, sonic anemometry and acoustic profiling in this environment. This combination allowed for unprecedented detail in observing the horizontal and vertical thermal and dynamic structure across the land-snow-air continuum. These data can be used to



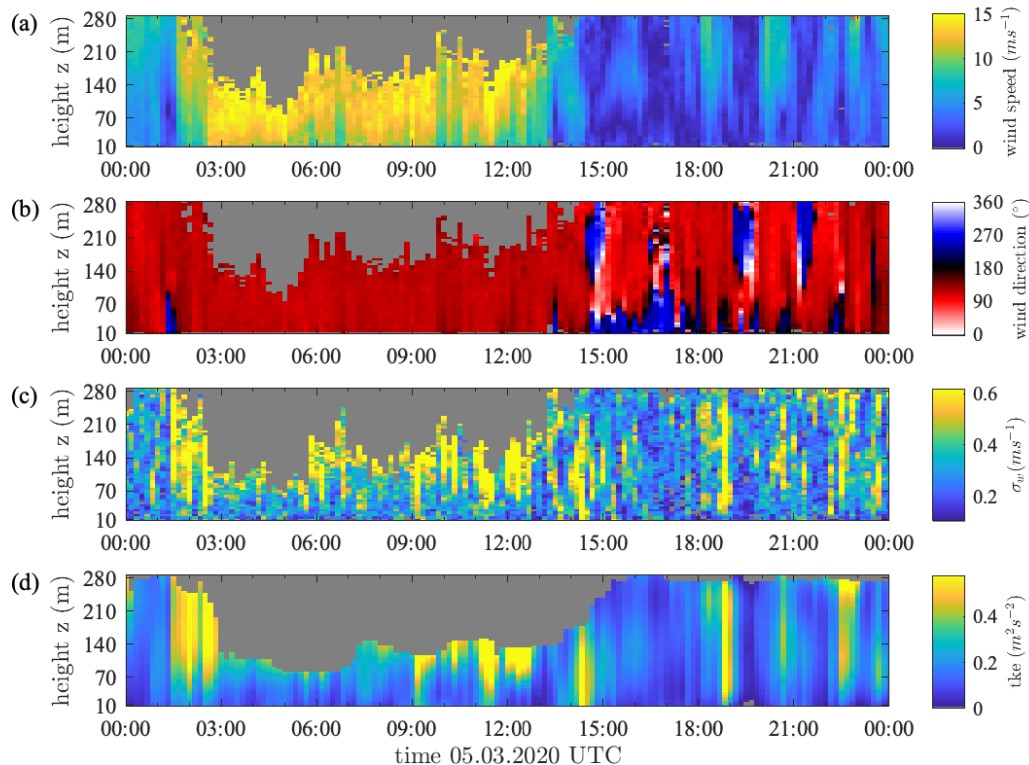

**Figure 8.** Time-height plots (sodargrams) observed with the miniSodar Data on 5th March 2020: (a) horizontal wind speed, (b) horizontal wind direction, (c) standard deviation of the vertical wind velocity $\sigma_w$, (d) turbulence kinetic energy (tke). Each pixel represents an average over the gate height of 5 m and period of 10 min. Missing data are displayed grey.

explore the role of submeso-scale motions on the SBL in the Arctic. Our complementary setup provides the unique ability to observe the role of topography on processes such as the interactions between katabatic outflows from the surrounding glaciers and the synoptic-scale flow over the Svalbard archipelago. The data allows for identifying the horizontal and vertical scales of turbulent and submeso-scale structures and their trajectories. First interpretation of findings support the dominant role of any topographic variation from decimeter to hundreds-of-meters scale on airflow and near-surface transport when flows and

turbulent transport are sufficiently weak, solely enabled by distributed sensing.

     One goal of the NYTEFOX campaign was to provide a proof-of-concept for future applications of horizontal fiber-optic distributed sensing technique in similarly challenging environments. While the feasibility of FODS has been proven for mid-latitude boundary layers (Thomas et al., 2012; Sayde et al., 2015; Peltola et al., 2020; Schilperoort et al., 2020; Pfister et al., 2021a), the high quality FODS data and its physical consistency with other more traditional near-surface meteorological ob-

servations, underline the technical feasibility and the functionality of FODS deployments in extreme temperature and wind



conditions of the polar regions. Note that temperatures during the measuring period dropped to $-30\,°\mathrm{C}$ with an average of $-17\,°\mathrm{C}$, which is extraordinarily cold for NY-Ålesund and more representative for the higher Arctic. This innovative observational technique therefore has unique merit to complement future boundary-layer studies, also in challenging environments, to observe motions and transport in a spatially resolving fashion across interfacial boundaries.

## 6   Data availability

The data availability from all observational systems for the NYTEFOX campaign period in February and March 2020 is summarized in Figure 9. Gaps in records were caused by instrument failure and post-field data processing described in section 4.1.

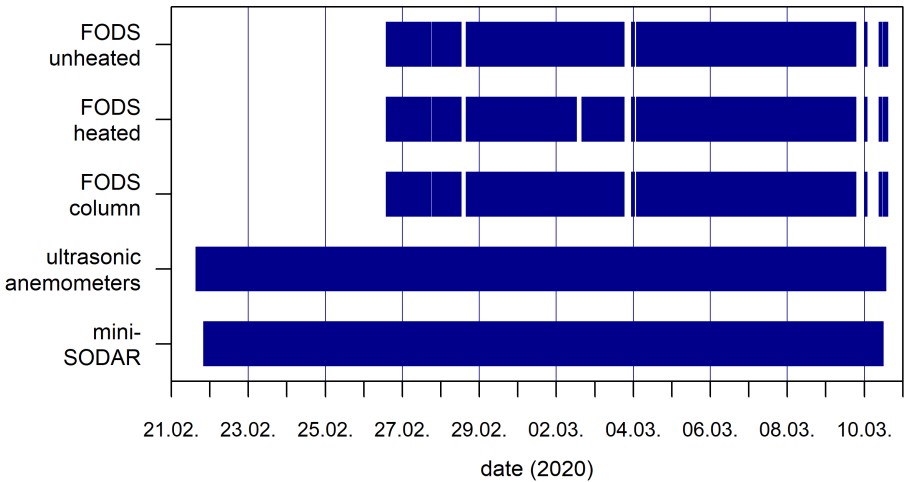

**Figure 9.** Data availability of the different data sets of NYTEFOX for the campaign period in February and March 2020.

The complete data set is available on Zenodo under the DOI:10.5281/zenodo.4335461.

*Author contributions.* Conceptualization & Methodology: CKT, AS, LP; Field data collection: JMH, MLZ, LP, DL, JS, AS, CKT; Software:
JMH, MLZ, KL, CKT; Scientific Data Analysis & Interpretation: MLZ, JMH, CKT; Data Curation: JMH, MLZ, AS, KL, CKT; Writing:
MLZ, JMH, AS, CKT; Visualization: MLZ, JMH, CKT; Supervision: CKT, AS; Project Administration: MLZ, JMH, LP, AS, CKT; Funding
Acquisition: JMH, MLZ, LP, AS, CKT

*Competing interests.* The authors declare no competing interests.





*Acknowledgements.* This project has received funding from the Alfred Wegener Institute for Polar and Marine Research (AWI) in Potsdam,
the European Research Council (ERC) under the European Union's Horizon 2020 research and innovation programme (grant agreement
no. 724629 DarkMix), and the Research Council of Norway (project number 291644) Svalbard Integrated Arctic Earth Observing System
(SIOS) – Knowledge Centre, operational phase. We thank the staff of the joint French-German AWIPEV-Station operated by the AWI and
the Polar Institute Paul Emile Victor (IPEV) in Ny-Ålesund and Irene Suomi (Finnish Meteorological Institute) for their great support.



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
