# Peer review of "NYTEFOX - The NY-Ålesund TurbulencE Fiber Optic eXperiment investigating the Arctic boundary layer, Svalbard"

_Earth System Science Data, 2021_

## Referee Comment (RC2)

Review of "NYTEFOX - The NY-Ålesund TurbulencE Fiber Optic eXperiment investigating the Arctic boundary layer, Svalbard" by Marie-Louise Zeller, Jannis-Michael Huss2, Lena Pfister, Karl E Lapo, Daniela Littmann, Johann Schneider, Alexander Schulz, and Christoph K Thomas

Ian Brooks

This paper documents a unque set of measurements obtained using a relatively new technique for temperature and wind measurement – fibre optic distributed sensing – along with some supporting measurements using well established techniques.

The installation is well documented, as are the data processing and quality control procedures. The example data and analysis shown is convincing as to the potential utility of the fibre optic measurement system for the study of small-scale, near-surface processes in the atmospheric boundary layer, particularly under conditions, as here, where spatial and temporal variability is high, and the statistical behaviour not necessarily stationary.

The data collection period is limited (2 weeks), and the location in NyÅlesund has bloundary-layer conditions strongly controlled by the local steep orography. The wider applicability of the data set for research purposes is thus perhaps limited; nevertheless it makes an excellent data set with which to gain practical experience of the fibre optic measurements, and assess its potential for other applications. This paper in particular is a valuable resource demonstrating the potential of this system.

The archived data on Zenodo is accompanied by appropriate documentation of both the installation and archived variables.

The paper is generally clear and well written, some minor editorial corrections are noted below.

**Editorial comments**

- Line 4: '...transport in temperatures, wind...' -> '...transport of temperature, wind...'
- Line 17: '...role in Earth system...' -> '...role in the Earth system...'
- Line 23: '...as the rate of warming affects the Arctic more than twice as fast as global average...' -> '...as the rate of warming in the Arctic is more than twice as fast as the global average...'
- Line 27: define acronym AWIPEV.
- Line 28: 'Located at 79°N in the Arctic ocean it...' -> 'Located at 79°N it...' Svalbard isn't really within the Arctic Ocean.
- Line 45: '....high temporal (9 s) and spatial (0.127 m) scales.' -> '...high temporal (9 s) and spatial (0.127 m) resolution.'
- Line 55: 'to close the observational gap between point measurements of the operational infrastructure at AWIPEV...' -> "to close the observational gap between point measurements made by the operational infrastructure at AWIPEV...'
- Line 63: Give the dates of the measurement campaign along with the location here these are only given in the abstract, nowhere in the main text.
- Lines 88-100: In the discussion of the installed equipment and figures 2 and 3 please provide information on all the items labelled 'a', 'b' etc on the figures. Some are documented here, but

not all...for example, what is the item labelled '(a) device' on figure 2? It would be useful if these were also all listed briefly in the caption to figure 2 for easy reference.

- Line 161: '(referred as low-resolution...' -> '(referred to as low-resolution...'
- Line 253: '...temperature instationarities...' -> '...temperature heterogeneity...' or '...temperature nonstationarity...' depending on the precise meaning intended – spatial differences or changing statistical properties either spatially or temporally.

Line 259: 'The visualization proofs...' -> 'The visualization proves...'

---

## Author Comment (AC1)

Final response to the reviews of 'NYTEFOX - The NY-Ålesund TurbulencE Fiber Optic eXperiment investigating the Arctic boundary layer, Svalbard' on behalf of all authors

Dear referees,

thank you for your thorough reviews helping to improve our manuscript! We considered all requested changes very useful and implemented the following revisions:

Responding to **referee #1**, we added three columns to Table 1, stating the accuracy of each quantity observed by the respective observing system as follows:

**Table 1.** Specifications of the measurement techniques: sampling rate, temporal resolution and averaging, spatial averaging and accuracy of the measurements. The accuracy of the MiniSodar and CSAT3 wind measurements were taken from their manuals. The accuracy of the CSAT3 temperature measurements was calculated by Fritz et al. (2021). Accuracy of temperature measured by FODS is based on the readings in the calibration baths: the bias is defined as the standard deviation of the daily averaged differences between the fiber- and reference (PT100) temperatures in each bath; the precision is defined as the median of the daily spatial standard deviation of fiber temperatures within each bath. The accuracy of the fiber wind speed is computed as the standard deviation of the fractional absolute deviation of the fiber readings from those of the reference (CSAT3) instrument aggregated to $30\,\mathrm{s}$. The bias depends systematically on location along the fiber and ranges from $8\,\%$ underestimation to $13\,\%$ overestimation with an average overestimation of $4\,\%$.

| measurement | sensing direction | sampling rate (s) | temporal resolution (s) | temporal averaging (s) | spatial averaging (m) | temp. accuracy (K) | wind speed accuracy | wind dir. accuracy (°) |
|---|---|---|---|---|---|---|---|---|
| metal encased fiber | horiz/vert | 3 | 9 | 6 | 0.127 | bias: 0.04 precision: 0.36 | 16 % | - |
| PVC fiber | vertical | 3 | 9 | 3 | 0.30 | bias: 0.03 precision: 0.28 | - | - |
| CSAT3 | - | 0.05 | - | 120 600 | 0.116 | <0.2 | 2 to 6 % | 0.7 |
| miniSodar | vertical | 10 | - | 600 | 5.0 | - | $0.1$ to $0.3\,\mathrm{m\,s^{-1}}$ | <1.5 |

In response to the review by **Ian Brooks** the following editorial changes were made:

- Line 4: '... transport in temperatures, wind...' -> '...transport of temperature, wind...'
- Line 17: '...role in Earth system...' -> '...role in the Earth system...'
- Line 23: '...as the rate of warming affects the Arctic more than twice as fast as global average...' ->'...as the rate of warming in the Arctic is more than twice as fast as the global average...'
- Line 27: AWIPEV Station -> 'joint French-German AWIPEV-Station operated by the Alfred Wegener Institute (AWI) and the Polar Institute Paul Emile Victor (IPEV)'
- Line 28: 'Located at 79°N in the Arctic ocean it...' -> 'Located at 79°N it...'
- Line 45: '...high temporal (9 s) and spatial (0.127 m) scales.' -> '...high temporal (9 s) and spatial (0.127 m) resolution.'
- Line 55: 'to close the observational gap between point measurements of the operational infrastructure at AWIPEV...' -> 'to close the observational gap between point measurements made by the operational infrastructure at AWIPEV...'
- Line 63: 'The experiment was conducted in Ny-Ålesund (78°55'24''N,11°55'15''O), one of the ...' -> 'The experiment was conducted for a period of 14 days from 26 February to 10 March 2020 in Ny-Ålesund (78°55'24''N,11°55'15''O). Ny-Ålesund is one of the ...'
- Lines 88-100: Caption figure 2: 'Schematic setup and picture of setup from the Zeppelin mountain from south. The fiber-optic array has a length of 700m. The letters a-i refer to the same elements.' → 'Schematic setup and picture of setup from the Zeppelin mountain from south. The fiber-optic array has a length of 700m. The letters refer to the locations of a: FODS device; b, c, g, h: road crossings; d, e, i: 10m towers and f: MiniSodar. The letters a-i refer to the same elements in both pictures.'
- Line 161: '(referred as low-resolution...' -> '(referred to as low-resolution...'
- Line 253: '...temperature instationarities...' → 'After the breakdown of the strong winds around 13:15 UTC and first meandering at this time it takes almost two hours for the temperature to drop and characteristic weak wind, submeso-scale structures to dominate. The latter appear as oscillating wind direction and strong temperature non-stationarities which start around 15 UTC.'
- Line 259: 'The visualization proofs...' -> 'The visualization proves...'

As requested by the **topical editor**, the DOI, referring to the publicly available dataset on Zenodo was replaced ('Abstract') and supplemented ('Data availability') by the full link, simplifying the accessibility for the reader.